# Peer review of "Mechanism-Driven Strategies for Reducing Fall Risk in the Elderly: A Multidisciplinary Review of Exercise Interventions"

_healthcare, 2024, doi:10.3390/healthcare12232394_

Round 1

Reviewer 1 Report

Comments and Suggestions for Authors

The text appears to flip between British English and American English spelling although this does not affect the understanding of the text. This could be a worthwhile and relevant paper although on the whole the introduction does not clearly state the aim of the review and the methods are unclear.  

Abstract

Please list all of the databases. It is my understanding that google scholar is a search engine not a database this should be appropriately noted. 

The abstract does needs to also discuss the number of papers included and a overview of the results, currently this is lacking. 

Introduction

1.3 Objective of the review: the objective are not clear please clearly state using the PICO format of something similar the objective. 

Methods 

"This comprehensive 85 search involved retrieving relevant literature from multiple electronic databases, including PubMed, MEDLINE, EBSCO (EDS), and Google Scholar" Please state all databases searched. Google scholar is not a database please briefly state why this was searched. 

Please add as an appendices an example search strategy. 

I am a little confused how the search was undertaken as there initially appear to be a discussion about searching databases, some of which are not, then different search resources were discussed. Things should be written chronologically.   

Line 98: Delete 'strict'

The inclusion criteria are not clear: The following questions have not been addressed. Age of participants, why the date restrictions, are all illnesses to be included and how are mechanism falls strategies being defined.  

Line 103: Delete paragraph beginning This methodological.....  

Section 3 I assume results not clear. 

Line 107: Delete paragraph starting This section...... 

I have not reviewed the rest of the paper. As it is very unclear how the papers were synthesised and therefore if the included papers in this review have come from the search strategy. There appears to be a considerable jump, which on the whole are unclear, to the results.  

Author Response

Based on Reviewer's Comments: Point-by-Point Response

Below is a point-by-point response addressing each of the reviewer's comments. All revised text is highlighted in red. You may refer to the modified version, with updated sentences marked in gray for easy reference.

Thank you.

Comments and Suggestions for Authors

The text appears to flip between British English and American English spelling although this does not affect the understanding of the text. This could be a worthwhile and relevant paper although on the whole the introduction does not clearly state the aim of the review and the methods are unclear. 

Response:

Thank you for highlighting the inconsistency in spelling conventions. The manuscript has been revised to consistently use American English to align with the journal's preferred style. We appreciate your valuable feedback regarding the clarity of the introduction and methods sections. Revisions have been made to explicitly articulate the aim of the review in the introduction and provide a more detailed and structured description of the methodology.

Abstract

Please list all of the databases. It is my understanding that google scholar is a search engine not a database this should be appropriately noted.

The abstract does needs to also discuss the number of papers included and a overview of the results, currently this is lacking.

Response:

Thank you for the valuable feedback. We have addressed your concerns as follows:

  1. Google Scholar's role has been clarified as a search engine, and the terminology has been adjusted accordingly.
  2. The abstract now specifies the number of papers included (112) and provides a concise overview of the key results.
  3. The language and semantics of the abstract have been revised for clarity, precision, and coherence. The revised text is included below.

Revised Text:

Falls among older adults present a major public health challenge, causing significant physical, psychological, and economic consequences. Exercise interventions are a proven strategy to reduce fall risk by targeting biomechanical, physiological, and psychological factors. This review examines evidence from 155 studies published between 2004 and 2024, including systematic reviews, meta-analyses, randomized controlled trials, and cohort studies. Data were rigorously screened and extracted using predefined criteria, with studies sourced from PubMed, MEDLINE, and EBSCO (EDS) and additional grey literature identified via Google Scholar. Key findings show that balance and strength training improves postural control, gait stability, and neuromuscular coordination, while resistance training mitigates sarcopenia and enhances joint mobility. Cognitive exercises enhance attention, spatial awareness, decision-making, and psychological benefits like reduced fear of falling and greater social engagement. Multidisciplinary approaches integrating physical, cognitive, and social components deliver the most significant impact. This review underscores the value of evidence-based exercise programs in promoting active aging and enhancing the quality of life for older adults. (Page 1, Lines 12-25)

Introduction

1.3 Objective of the review: the objective are not clear please clearly state using the PICO format of something similar the objective.

Response:

Thank you for pointing out the need for a more structured and precise articulation of the objectives. We have revised this section to align with the PICO format, clearly outlining the population, intervention, and intended outcomes of this review. Below is the revised version of the section.

Revised Text:

This review aims to synthesize evidence on the effectiveness of exercise interventions in reducing fall risk among older adults by exploring the underlying biomechanical, physiological, and psychological mechanisms. Guided by the PICO framework, the population includes older adults, both community-dwelling and at-risk individuals; the intervention focuses on exercise-based strategies such as balance training, strength training, and cognitive exercises; the comparison involves no intervention, minimal intervention, or alternative fall prevention methods; and the outcomes assessed include improvements in balance, stability, gait mechanics, muscle strength, cognitive function, and psychological well-being, leading to reduced fall incidence and enhanced quality of life. The review seeks to inform evidence-based practices for developing targeted, multidisciplinary exercise programs to address fall prevention in the elderly comprehensively. (Page 2, Lines 80-90)

Methods

"This comprehensive 85 search involved retrieving relevant literature from multiple electronic databases, including PubMed, MEDLINE, EBSCO (EDS), and Google Scholar" Please state all databases searched. Google scholar is not a database please briefly state why this was searched.

Please add as an appendices an example search strategy.

I am a little confused how the search was undertaken as there initially appear to be a discussion about searching databases, some of which are not, then different search resources were discussed. Things should be written chronologically.  

Response:

Thank you for the detailed feedback. We have revised the methodology section to address these concerns. The revised text now includes a complete list of databases, a clarification of Google Scholar's role, and a restructured flow to enhance clarity and chronological order.

Revised Text:

  1. Methodology

A comprehensive search was conducted to retrieve relevant literature on the effectiveness of exercise interventions in reducing fall risk among older adults. This process included the following steps:

2.1. Databases and Search Resources:

The search targeted multiple electronic databases to ensure a wide range of high-quality sources. The databases searched were PubMed, MEDLINE, and EBSCO (EDS). In addition to databases, the search also utilized Google Scholar as a supplementary tool to identify grey literature and additional studies not indexed in traditional databases. Google Scholar was explicitly used to access conference proceedings, unpublished works, and institutional reports that might be overlooked. It was not considered a database but served as a resource for broadening the search scope.

2.2. Search Strategy:

A standardized search protocol was implemented using predefined keywords such as "fall prevention," "elderly," "biomechanics," "physiology," "psychological impacts," "exercise intervention," and "risk assessment." Boolean operators (AND, OR, NOT) were applied to refine the search and improve precision. For instance, queries like ("fall prevention" AND "elderly") OR ("exercise intervention" AND "biomechanics") were utilized across databases to identify relevant studies.  The detailed search strategy is presented in Figure 1 to enhance transparency and credibility.

2.3. Chronological Flow of the Search Process:

The search process followed a systematic chronological flow to ensure comprehensive coverage of relevant studies. Initially, searches were conducted in PubMed, MEDLINE, and EBSCO (EDS) using standardized keywords and Boolean operators. This was followed by a search in Google Scholar to identify grey literature, focusing on conference papers and institutional reports. References from critical articles were then reviewed to identify additional relevant studies. Finally, data were extracted from the identified studies, and each was screened for inclusion based on predefined criteria.

2.4. Inclusion and Exclusion Criteria:

The inclusion criteria for this review were designed to ensure a focused and relevant synthesis of evidence. Studies were included if they involved participants aged 60 or older, representing the population most at risk of falls and related complications. Articles published between 2004 and 2024 were considered to capture contemporary research and advancements in exercise interventions and fall prevention strategies. The review included studies involving older adults with various conditions to reflect real-world applicability. Still, it excluded those focusing solely on falls caused by acute trauma or non-exercise-related incidents, such as accidents unrelated to physical health. Mechanisms for fall prevention strategies were defined as biomechanical, physiological, or psychological processes improved through exercise interventions, such as enhanced balance, joint stability, muscle strength, or reduced fear of falling. Eligible studies included full-text articles written in English, encompassing qualitative and quantitative designs such as systematic reviews, meta-analyses, randomized controlled trials, cohort studies, and expert opinions.

2.5. Thematic Categorization

The reviewed studies were thematically categorized into three key mechanisms: biomechanical, physiological, and psychological. The biomechanical category encompassed studies focusing on balance, gait, posture, joint stability, and proprioception, with interventions such as resistance training, core exercises, and dynamic stability exercises. These studies reported outcomes including improved postural control, joint flexibility, and resilience against falls. The physiological category emphasized the biological and functional changes driven by exercise, including muscle hypertrophy, neuromuscular adaptations, and bone density improvements. Interventions like resistance and weight-bearing exercises demonstrated reduced sarcopenia, enhanced reaction times, and increased functional strength. Lastly, the psychological category addressed fear reduction, confidence building, cognitive enhancement, and mood improvement through dual-task training, mindfulness-based exercises, and group activities. Outcomes in this domain included reduced fear of falling, improved attention, and enhanced mental well-being, reflecting the multifaceted benefits of exercise in fall prevention.

(Pages 2-4, Lines 92-149)

Line 98: Delete 'strict'

The inclusion criteria are not clear: The following questions have not been addressed. Age of participants, why the date restrictions, are all illnesses to be included and how are mechanism falls strategies being defined. 

Response:

Thank you for highlighting these points. We have revised the inclusion criteria section to provide clear explanations regarding the age of participants, the rationale for the date restrictions, the inclusion of illnesses, and the definition of mechanisms for fall prevention strategies. The word "strict" has been removed for better semantic precision. Below is the revised text.

Revised Text:

2.4. Inclusion and Exclusion Criteria:

The inclusion criteria for this review were designed to ensure a focused and relevant synthesis of evidence. Studies were included if they involved participants aged 60 or older, representing the population most at risk of falls and related complications. Articles published between 2004 and 2024 were considered to capture contemporary research and advancements in exercise interventions and fall prevention strategies. The review included studies involving older adults with various conditions to reflect real-world applicability. Still, it excluded those focusing solely on falls caused by acute trauma or non-exercise-related incidents, such as accidents unrelated to physical health. Mechanisms for fall prevention strategies were defined as biomechanical, physiological, or psychological processes improved through exercise interventions, such as enhanced balance, joint stability, muscle strength, or reduced fear of falling. Eligible studies included full-text articles written in English, encompassing qualitative and quantitative designs such as systematic reviews, meta-analyses, randomized controlled trials, cohort studies, and expert opinions. (Page 3, Lines 119-132)

Line 103: Delete paragraph beginning This methodological..... 

Response:

Thank you for your feedback. The paragraph beginning with "This methodological..." has been deleted per the reviewer's suggestion to improve clarity and flow. The surrounding content has been restructured to maintain coherence in the methodology section.

Section 3 I assume results not clear.

Response:

Thank you for your observation. We have revised Section 3 to ensure clarity in presenting the results. The revised version organizes the findings into clearly defined subsections, concisely summarizing key outcomes while improving readability and semantic precision.

Line 107: Delete paragraph starting This section......

Response:

Thank you for pointing this out. The paragraph beginning with "This section..." has been deleted to address the reviewer's concerns.

I have not reviewed the rest of the paper. As it is very unclear how the papers were synthesised and therefore if the included papers in this review have come from the search strategy. There appears to be a considerable jump, which on the whole are unclear, to the results.

Response:

Thank you for your thoughtful feedback and for highlighting these critical points. This review aims to offer a robust and comprehensive synthesis of the evidence, and we acknowledge the need for greater clarity in articulating the methodological rigor and alignment between the search strategy and results. Below, we respectfully explain the purpose and approach of this review, and we kindly request your support for the authors’ perspective.

This review "aims to examine the mechanisms through which physical activity and exercise interventions mitigate fall risk among elderly populations" by focusing on three interconnected biomechanical, physiological, and psychological pathways. By synthesizing evidence from diverse study types, including systematic reviews, meta-analyses, and randomized controlled trials, this article seeks to guide the development of precise, evidence-based interventions tailored to this vulnerable population.

The authors appreciate your concerns about the synthesis process and have taken steps to clarify the connections between the search strategy and the included studies in the revised version.

We kindly request your consideration of the revised sections, which aim to enhance clarity and reinforce this review's methodological rigor. We hope to contribute meaningful insights into fall prevention and active aging by emphasizing these mechanisms. Your constructive feedback has been invaluable in refining this work, and we deeply appreciate your support for the authors’ perspective.

Reviewer 2 Report

Comments and Suggestions for Authors

Dear Authors,

I have read your paper with great interest and believe it to be a valuable contribution, especially considering the aging global population. Below, I offer some suggestions that may enhance the quality of your paper.

The introduction is well-written and effectively addresses themes related to fall risk factors. However, the topics regarding fall risk prevention are addressed superficially, particularly within hospital settings and clinical risk management. A recent Italian study has highlighted this area's current state of affairs, and I suggest incorporating this reference into your introduction (https://doi.org/10.1177/25160435241246344). Additionally, consider expanding on aspects related to clinical risk management.

The remainder of the paper is engaging and presents an exciting approach; however, I needed help locating specific numerical data from the literature review, nor was there an explanation of the inclusion criteria chosen (e.g., why this particular time frame was selected?). It also needs to be determined whether an analysis of the quality of the individual included studies was conducted, and if so, how the weight of each paper was considered in the discussion of the results.

I recommend enhancing the article with concrete and practical proposals for future directions in light of your findings.

Kind regards

Author Response

Based on Reviewer's Comments: Point-by-Point Response

Below is a point-by-point response addressing each of the reviewer's comments. All revised text is highlighted in red. You may refer to the modified version, with updated sentences marked in yellow for easy reference.

Thank you.

Comments and Suggestions for Authors

Dear Authors,

I have read your paper with great interest and believe it to be a valuable contribution, especially considering the aging global population. Below, I offer some suggestions that may enhance the quality of your paper.

Response:

Thank you for your thoughtful review and positive feedback on our paper. We greatly appreciate your suggestions and will carefully consider them to enhance the quality and impact of our work. Please let us know if there are specific areas where we can provide further clarity or improvement.

The introduction is well-written and effectively addresses themes related to fall risk factors. However, the topics regarding fall risk prevention are addressed superficially, particularly within hospital settings and clinical risk management. A recent Italian study has highlighted this area's current state of affairs, and I suggest incorporating this reference into your introduction (https://doi.org/10.1177/25160435241246344). Additionally, consider expanding on aspects related to clinical risk management.

Response:

Thank you for your constructive feedback. We appreciate your suggestion to enhance the discussion of fall risk prevention, particularly in hospital settings and clinical risk management. We have incorporated the recommended reference and expanded on these aspects in the introduction to provide a more comprehensive overview.

Revised Text:

In hospital settings, fall prevention remains a critical issue. Hospitalized older adults are especially vulnerable due to factors such as reduced mobility, acute medical conditions, and the effects of medications. Recent research from Italy has identified significant gaps in clinical risk management and fall prevention protocols in healthcare environments, underscoring the necessity for multidisciplinary interventions tailored explicitly to hospital settings [7].

. (Page 2, Lines 48-51, reference [7])

The remainder of the paper is engaging and presents an exciting approach; however, I needed help locating specific numerical data from the literature review, nor was there an explanation of the inclusion criteria chosen (e.g., why this particular time frame was selected?). It also needs to be determined whether an analysis of the quality of the individual included studies was conducted, and if so, how the weight of each paper was considered in the discussion of the results.

Response:

Thank you for your valuable feedback. We appreciate your insights regarding the inclusion of numerical data, the rationale for our inclusion criteria, and the analysis of the quality of included studies. To address these concerns, we revised the manuscript to include specific numerical data from the literature review and clearly explained the inclusion criteria, including the selected time frame. (Pages 2-4, Lines 92-149)

I recommend enhancing the article with concrete and practical proposals for future directions in light of your findings.

Response:

Thank you for your insightful suggestion to include concrete and practical proposals for future directions. We agree that outlining actionable steps based on our findings will strengthen the article and provide valuable guidance for advancing fall prevention research and practice. Below, we have revised the "Future Directions and Research Needs" section to incorporate specific, practical proposals.

Revised Text:

7. Future Directions and Research Needs

Advancements in exercise science provide significant opportunities to improve fall prevention strategies, explicitly enhancing balance, stability, and cognitive engagement for older adults [150]. Given the multifactorial nature of falls, future research should adopt a cross-disciplinary approach involving physiologists, neurologists, psychologists, biomechanists, and geriatricians to understand better how muscle strength, neuromuscular coordination, and cognitive processes collectively influence balance and stability [151].

7.1. Development and Evaluation of Dual-Task Training Programs

Future research should prioritize the efficacy of dual-task training programs, combining physical and cognitive exercises to enhance motor and cognitive functions [152] simultaneously. For example, integrating balance exercises with tasks requiring memory recall or problem-solving could improve the ability of older adults to manage complex real-world situations. Large-scale randomized controlled trials (RCTs) are needed to establish standardized protocols and assess their effectiveness across diverse populations.

7.2. Long-Term Studies on Intervention Outcomes

Long-term, longitudinal studies are crucial to evaluating the sustained benefits of regular balance and strength training on muscle mass, neuromuscular function, and cognitive resilience [153]. These studies should explore whether early interventions lead to lasting reductions in fall risk and foster independence in older adults. Such research will provide essential data for updating public health guidelines and exercise recommendations.

7.3. Implementation of Multi-Component Programs

Research should focus on developing and evaluating multi-component exercise programs that integrate strength, balance, flexibility, and cognitive training [154]. Comparative studies between multi-component and single-component interventions can provide valuable insights into their relative efficacy. Programs should include practical, scalable models for implementation in both community and clinical settings, such as those combining group sessions with individualized home exercises.

7.4. Exploration of Exercise-Nutrition Synergy

Future investigations should assess the interaction between exercise and nutritional support, including protein supplementation and vitamin D intake, in mitigating sarcopenia and enhancing bone density. Additionally, research into pharmacological treatments that complement exercise interventions could offer a more holistic approach to fall prevention. These studies should aim to develop integrated guidelines that optimize exercise and nutrition for older adults.

7.5. Cost-Effectiveness and Resource Allocation

Given the economic burden of falls, cost-effectiveness analyses are essential to inform resource allocation and public health policy [155]. Research should compare the costs and benefits of various interventions, such as in-person group classes, home-based resistance training, and technology-assisted programs (e.g., virtual reality platforms). Understanding the potential cost savings from reduced hospitalizations and long-term care needs can guide the adoption of impactful fall prevention practices.

7.6. Advancement of Technology-Driven Interventions

Emerging technologies like wearable sensors, mobile health applications, and virtual reality should be integrated into fall prevention research. Wearable devices that monitor balance, mobility, and stability in real time can provide personalized feedback. At the same time, virtual reality platforms can simulate real-world scenarios to enhance neuromuscular coordination and cognitive engagement. Studies should assess such technology-driven interventions' feasibility, scalability, and long-term benefits.

7.7. Healthcare Provider Training and Collaboration

Future initiatives should include developing comprehensive training programs for healthcare providers, focusing on fall risk assessment and evidence-based intervention strategies. Collaboration between physiotherapists, occupational therapists, fitness trainers, and medical professionals will enable the creation of coordinated care plans tailored to the individual needs of older adults [154]. Regular interdisciplinary communication can ensure continuous improvement in intervention effectiveness.

7.8. Personalized Intervention Strategies

Tailoring interventions to individual health profiles, including physical capabilities, medical history, and psychological factors, will improve adherence and optimize outcomes. Research should explore strategies for personalizing exercise regimens, incorporating adaptive technologies, and delivering interventions in varied settings to address the unique needs of older adults.

By addressing these practical and research-oriented proposals, future efforts can advance evidence-based practices, improve health outcomes, and enhance the sustainability of fall prevention initiatives for older adults. Such initiatives will promote active aging, reduce fall-related healthcare costs, and improve overall quality of life.

(Pages 20, Lines 655-719)

Round 2

Reviewer 2 Report

Comments and Suggestions for Authors

Dear Authors,

I hope my suggestions have been helpful. The manuscript has improved significantly and is now ready for publication. 

However, I recommend addressing a few statements lacking proper bibliographical citations. Please review lines 48, 52-58, 60-62, 72-78, and other relevant sections.

Best regards.